# Size Effect of Graphene Quantum Dots on Photoluminescence

**DOI:** 10.3390/molecules26133922

**Published:** 2021-06-26

**Authors:** Ziyi Liu, Fei Li, Yi Luo, Ming Li, Guanghui Hu, Xianjuan Pu, Tao Tang, Jianfeng Wen, Xinyu Li, Weitao Li

**Affiliations:** 1College of Science, Guilin University of Technology, Guilin 541004, China; liuziyi0917@163.com (Z.L.); m17753101516@163.com (F.L.); adam2513@163.com (Y.L.); hu-guanghui-rm@ynu.jp (G.H.); tangtao@glut.edu.cn (T.T.); wjfculater@163.com (J.W.); lixinyu5260@163.com (X.L.); 2Shanghai Applied Radiation Institute, School of Environmental and Chemical Engineering, Shanghai University, Shanghai 200444, China; 3Textile and Garment Industry of Research Institute, Zhongyuan University of Technology, Zhengzhou 450007, China

**Keywords:** graphene quantum dots, size effect, photoluminescence

## Abstract

High-photoluminescence (PL) graphene quantum dots (GQDs) were synthesized by a simple one-pot hydrothermal process, then separated by dialysis bags of different molecular weights. Four separated GQDs of varying sizes were obtained and displayed different PL intensities. With the decreasing size of separated GQDs, the intensity of the emission peak becomes much stronger. Finally, the GQDs of the smallest size revealed the most energetic PL intensity in four separated GQDs. The PL energy of all the separated GQDs shifted slightly, supported by density functional theory calculations.

## 1. Introduction

Recently, luminescent carbon materials have received considerable attention, including carbon nanotube [1,2], carbon quantum dots [3,4], reduced graphene oxide quantum dots [5], graphene oxide [6], and graphene quantum dots (GQDs) [7]. Compared with traditional semiconductor quantum dots, such as CdX (X = S, Se) and PbS, GQDs provide various advantages, such as more environment-friendly, low-cost, easy-to-prepare, relatively stable chemical internees and strong luminescence. In particular, GQDs, as a zero-dimension graphene material, has the characteristics of graphene and unique quantum confinement effects and edge effects [8,9,10]. These proprieties endow GQDs with stable fluorescence and adjustable bandgap [11,12,13]. Based on the above excellent feature, GQDs could be applied in multiple fields, such as photovoltaic devices [14], catalysis [15,16,17,18,19,20,21,22], solar cells [23,24], sensing [25,26], drug delivery [27], and cell imaging [28,29,30,31]. The hydrothermal method is a commonly used method for preparing GQDs. Pan et al. first prepared GQDs by chemically cutting graphene nanosheets through acid and a hydrothermal environment [10,32]. But the arrangement of the GQD prepared by this method is not regular. Dong et al. used the acid exfoliation method to prepare graphene quantum dots by chemically oxidizing carbon black (CB) in HNO_3_ under reflux conditions. However, it is difficult to remove excess oxidant (for example, HNO_3_) from the solution [33]. Various methods for preparing GQDs have been reported [34,35]. The fluorescences of GQDs synthesized by different methods mostly display blue, green, and yellow colors. Their photoluminescence (PL) properties have been studied [36,37]. To adjust the bandgap and electronic density effectively, strengthen chemical activity, increase the quantum yield (QY), and expand the practical application of GQDs, a large number of published research studies have focused on modifying GQDs by doping heteroatoms [38], for instance, nitrogen and sulfur [39], chlorine [40], and fluorine [7]. However, the effect of GQDs with different sizes on PL intensity has not been thoroughly investigated.

We found that there are few studies on whether the PL emission intensities of GQDs of different sizes are different. In order to fill the gap in this regard, we synthesized GQDs by a simple one-step hydrothermal method and then obtained four separated GQDs using dialysis bags with different molecular weights. The dimension changes and PL properties of the separated GQDs were investigated. Furthermore, the relationship between the size of the obtained GQDs and their PL energy shift is discussed. Overall, DFT calculation was used to illustrate the PL energy shift.

## 2. Experimental Section

### 2.1. Synthesis of GQDs

GQDs were prepared by a simple, green, bottom-up hydrothermal method [41]. Briefly, 1 g of pyrene was added to 80 mL of 65–68% nitric acid, then refluxed and stirred for 12 h at 80 °C. After cooling to room temperature, the product was diluted with 200 mL of distilled water (DI) and filtered by a 0.22 μm filter membrane to remove the excess acid, then dissolved in 150 mL of 0.0125 M aqueous NaOH solution by ultrasound to adjust the pH value to 11. Then the ultrasonic suspension was transferred to a Teflon autoclave and was heated at 180 °C for 12 h. Next, after cooling to room temperature, the obtained product was filtered through a 0.22 μm filter membrane to remove the insoluble product. To obtain the GQDs of different sizes, the GQD solution was separately dialyzed with 14,000, 7000, 3500, and 1000 Da molecular weight dialysis bags successively.

### 2.2. Computational Details

All calculations were performed using the Vienna Ab initio Simulation Package (VASP) [42,43]. The projector augmented-wave (PAW) potentials [44,45] with an energy cutoff of 800 eV were used for the plane-wave basis set. The generalized gradient approximation (GGA) in the form of the Perdew−Burke−Ernzerhof (PBE) functional was employed [46]. In this work, two different-sized systems of the armchair- and zigzag-edged GQDs were modeled (hexagonal cell). To minimize the artificial interactions due to periodic boundary conditions, each GQD was separated by a vacuum of ~15 Å. The Brillouin zone was sampled by 3 × 3 × 1 Monkhorst–Pack k-point grids. All atoms were relaxed until the residual forces on each atom were less than 0.01 eV/Å.

### 2.3. Characterization

Transmission electron microscopy (TEM) was carried out on a JEM-2100F (JEOL, Japan) electron microscope. Samples were prepared by placing a drop of dilute aqueous dispersion of GQDs on the surface of a copper grid. Powder X-ray diffraction (XRD) spectra were collected on a Rigaku D/Max 2550 diffractometer within 5°–80° (2 theta). X-ray photoelectron spectroscopy (XPS) measurements were made on Escalab 250XI with mono Al Kα radiation (hν = 1486.6 eV). Raman spectra were recorded on a laser confocal Raman spectrometer (Renishaw inVia) with 514 nm incident radiation. The photoluminescence spectra (PL) and photoluminescence excitation (PLE) spectra were recorded using a fluorescence spectrophotometer (Agilent Technologies, Cary Eclipse, Australia) at a sample concentration of 0.1 mg·mL^−1^. UV–VIS absorption spectra were recorded on a PerkinElmer Lambda 750 spectrophotometer at a sample concentration of 0.1 mg·mL^−1^. Fourier-transform infrared (FTIR) spectra were recorded using an IRAffinity-1S spectrometer (KBr pellet). Unless otherwise specified, the GQDs (the GQDs dialyzed with a dialysis bag of 1000 Da) with the best optical performance were selected for various characterizations.

## 3. Results and Discussion

The TEM and high-resolution TEM images of four separated GQDs dialyzed with dialysis bags of 14,000, 7000, 3500, and 1000 Da molecular weights are shown in Figure 1a–h. The separated GQDs are well dispersed with a uniform size distribution, whose average lateral sizes are about 10.33, 9.33, 8.42, and 6.53 nm, respectively (insets in Figure 1a–h). This indicates that the size of the separated GQDs obtained by dialysis with different molecular weights of dialysis bags is different. As the molecular weights of the dialysis bags decreased, the size of the obtained GQDs also reduced. High-resolution TEM images also show crystalline GQDs with a lattice measurement of 0.203 nm, which coincide with the graphene (002) plane [18]. The real-space images and their fast Fourier transform (FFT) patterns (insets in Figure 1e–h) with a hexagonal honeycomb structure demonstrate that the GQDs are nearly defect-free graphene single crystals.

To understand their structure, various spectral characterizations were used. Figure 2a shows XRD patterns of GQDs. We can observe a typical peak (002) at 2θ = 26.5°, and the interlayer spacing is 3.34 Å, which is identical to that of graphene [41,47]. FTIR spectroscopy was also revealed to be a further characteristic of samples. As shown in the FTIR spectra in Figure 2b, a distinct peak at 1590 cm^−1^ corresponds to the vibration of C=C bonds, and a broad vibration at about 3430 cm^−1^ for the O–H bonds [41,47,48]. The O–H peak is mainly given credit for the hydroxyl of the obtained GQDs and can be confirmed by the strong vibration of C=OH at about 1270 cm^−1^. The obvious peak at about 870 cm^−1^ is attributed to the vibration of C–H [48].

To further determine the component of GQDs, XPS measurement was employed. The XPS full spectra (Figure 3a) of the four separated GQDs obtained by different molecular weights of dialysis bags show that the strong peaks at approximately 284 eV (C 1s) and 533 eV (O 1s) exist in all GQDs, as well as a week peak from impurity Na^+^. The high-resolution spectrum of the C 1s reign of GQDs (Figure 3b) shows the strong peak of C=C at 284.8 eV and another distinguishable peak of COOH at 288.1 eV. The high-resolution O 1s spectrum (Figure 3c) reveals the existence of the peak of O–H at 531.4 eV. The high-resolution C 1s and O 1s spectra of the other three separated GQDs obtained by dialysis bags of 14,000, 7000, and 3500 Da molecular weights are shown in Appendix A. The data of XPS display that the obtained 1,3,6-trinitropyrene by water bath reaction synthesized GQDs by obliterating the NO_2_ group under a strong alkaline condition. The bonding position between the hydroxyl group and the single-crystalline GQD lattice is most likely at the edge rather than that at the basal plane. This unique edge functionalization property will not bring out any defects in the graphene basal plane, which is different from other functionalizations viewed in graphene oxide [41,49,50] and defective GQDs obtained by other methods [32,33,34,35,36,37,38,39,40,41,42,43,44,45,46,47,48,49,50,51,52].

For comparison, the Raman spectra of the four separated GQDs obtained with dialysis bags of different molecular weights were also measured. As shown in Figure 4, as the size of the GQD molecular decreases, the G and D peaks of the Raman spectrum become more and more obvious. Only GQDs obtained from dialysis bags with a molecular weight of 1000 Da have obvious G and D peaks. This is also closest to the characteristic values of a single-layer graphene. We generally use the ratio of peak D to peak G to indicate the density of defects in graphene. The greater the ratio, the greater the degree of defects, and vice versa. It can be seen from the Raman spectrum of the sample (Figure 4) that the G peak is always slightly higher than the D peak, so *I**_D_*/*I_G_* is less than 1, as shown in Table 1. This is consistent with what was mentioned in the previous article; that is, GQD is almost a defect-free single crystal of graphene.

To explore the optical properties of the four separated GQDs with varying sizes, the UV–VIS absorption and PL spectra were demonstrated. GQDs are highly soluble in DI water. The as-prepared GQD aqueous dispersion exhibited green fluorescence under irradiation with 365 nm UV light (inset in Figure 5a), which is consistent with the other three separated GQDs obtained by dialysis bags of 14,000, 7000, 3500 Da molecular weights (inset in Appendix A–c). As shown in Figure 5a, the GQDs exhibit two distinct excitonic absorption bands at about 365 and 490 nm in the UV–VIS absorption spectrum, which is similar to the GQDs synthesized by hydrothermal method [33]. The two absorption peaks can be assigned to the π–π* and n–π* transitions between the oxygen-/nitrogen-containing groups and sp^2^ domains [53,54]. As the size of GQDs increased, the intensity of the absorption peak became weaker (Figure 5a and Appendix A–c). This means that we can increase the absorption of ultraviolet light by reducing the size of the GQDs.

As displayed in Figure 5b and Appendix A–f, the maximum PL wavelengths of the four separated GQDs were non-change-excited at different wavelengths, exhibiting their excitation-independent property. When the four separate GQDs are irradiated with different wavelengths of ultraviolet light, the positions of their PL emission peaks are slightly different, which means that we can control the fluorescence color of the GQDs by changing the excitation wavelength. The emission peak of GQDs was observed at about 530 nm. Figure 5c shows the PL of GQDs of varying sizes under 330 nm. The results indicate that the PL intensities of the GQDs with different sizes were different. With the decreasing size of the GQDs, the PL intensity was enhanced. In these four separated GQDs, the GQDs on a dialysis bag of 1000 Da molecular weight have the strongest PL intensity, whereas the GQDs on a dialysis bag of 14,000 Da molecular weight has the weakest PL intensity. The PL excitation (PLE) spectrum (Figure 5d) of the GQDs after fixing the emission wavelength was shown at 530 nm, and one of the excitation peaks was shown at 384 nm, which was according to PL results. The PL disintegration curves show the single-exponential feature of the four separated GQDs (Appendix A). The fluorescent lifetime of the GQDs dialyzed with a dialysis bag of 1000 Da molecular weight was shortest to 1.88 ns, which is according to their smallest size.

To further investigate the optical properties of the GQDs obtained by dialysis bags of different molecular weights, the fluorescence QY measurements were calculated by the result of the PL emission intensity and absorption. Table 2 reveals the QY of the four separated GQDs. The results of the samples were calculated using the following formula: φ = φ_r_ (I/I_r_)(n^2^/n_r_^2^)(A_r_/A) [55], where φ represents fluorescence QY, I represents the integrated emission intensity, n shows the refractive index of the solvent (1.33 of water), A means optical density, and r stands for reference. The QY of the GQDs of 1000 Da molecular weight reached 0.45, which is higher than the others, and the result corresponds to that in Figure 5c. Combining the above data and analysis, we speculate that the number of defects contributes to the intensity of luminescence [18].

Figure 5c shows that the PL emission peaks of the four separated GQDs are slightly red-shifted. We believe that GQDs of different sizes have different band gap widths, resulting in different PL emission peaks. In order to prove this relationship, we performed DFT calculations. The results are illustrated in Figure 6. With the increasing size of the four separated GQDs, the gap of both the armchair- and zigzag-edged GQDs decreases rapidly at first and becomes gentle when the size is greater than 6 nm. When the size is greater than 5 nanometers, the curve begins to become flat. It is known that DFT–GGA methods often underestimate the gap, so we assume that with either an armchair or zigzag edge, the bandgap of GQDs changes lightly depending on their size when the size is greater than 5 nm. It can be seen from this that when the size of the GQD is larger, its band gap becomes smaller, which is consistent with the experimental results. As shown in Figure 5c, under excitation at 330 nm, the PL emission peaks of the four separate GQDs are slightly red-shifted as the GQD size increases. As the size of the GQD increases, its energy gap continues to decrease, and photons are more likely to transition, which results in a red shift of the PL emission peak. This result also confirms our conjecture.

## 4. Conclusions

In summary, we synthesized GQDs by a simple one-step hydrothermal method and then obtained different sizes of GQDs using dialysis bags with different molecular weights. The average lateral sizes of the GQDs obtained by dialysis bags of 14,000, 7000, 3500, 1000 Da molecular weights are about 10.33, 9.33, 8.42, and 6.53 nm, respectively. This indicates that the sizes of the GQDs obtained by dialysis with different molecular weights of dialysis bags are different. Meanwhile, the four separated GQDs displayed different PL intensities. With the decreasing size of the separated GQDs, the intensity of the emission peak becomes much stronger. The GQDs with the smallest size revealed the most energetic PL intensity in the four separated GQDs. DFT calculations show that with a size greater than 6 nm, the bandgap of the GQDs changes slightly. This is consistent with the experimental results.

## Figures and Tables

**Figure 1 molecules-26-03922-f001:**
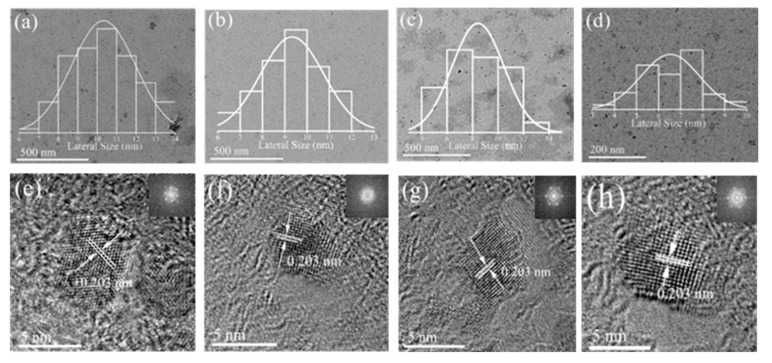
TEM images of four separated GQDs dialyzed with dialysis bags of 14,000 Da (**a**), 7000 Da (**b**), 3500 Da (**c**), and 1000 Da (**d**) molecular weights; insets are corresponding lateral size distributions. High-resolution TEM images of four separated GQDs dialyzed with dialysis bags of 14,000 Da (**e**), 7000 Da (**f**), 3500 Da (**g**), 1000 Da (**h**) molecular weights; insets are corresponding FFT patterns.

**Figure 2 molecules-26-03922-f002:**
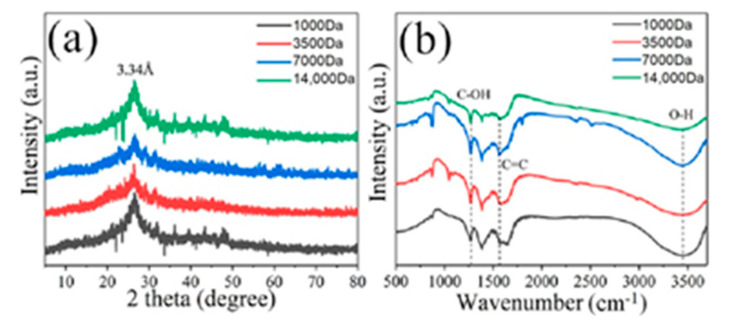
XRD patterns (**a**) and FTIR spectra (**b**) of four separated GQDs.

**Figure 3 molecules-26-03922-f003:**
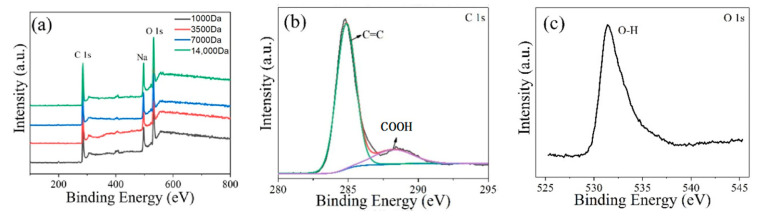
(**a**) XPS full spectra of four separated GQDs. High-resolution C 1s (**b**) and O 1s spectra (**c**) of GQDs dialyzed with a dialysis bag of 1000 Da molecular weight.

**Figure 4 molecules-26-03922-f004:**
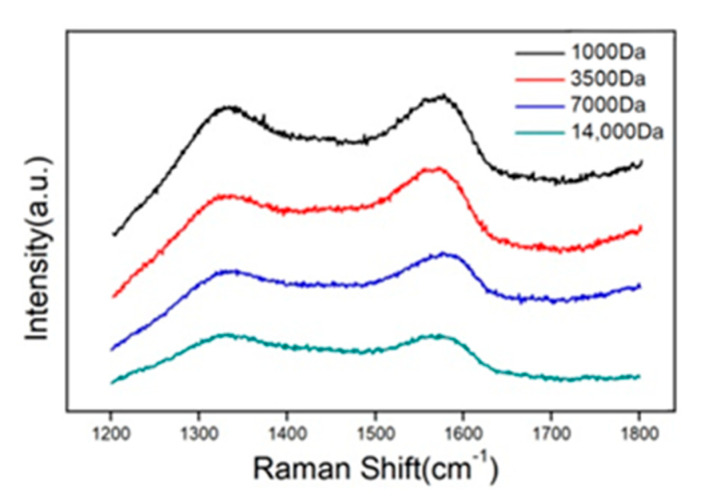
Raman spectra of the four separated GQDs.

**Figure 5 molecules-26-03922-f005:**
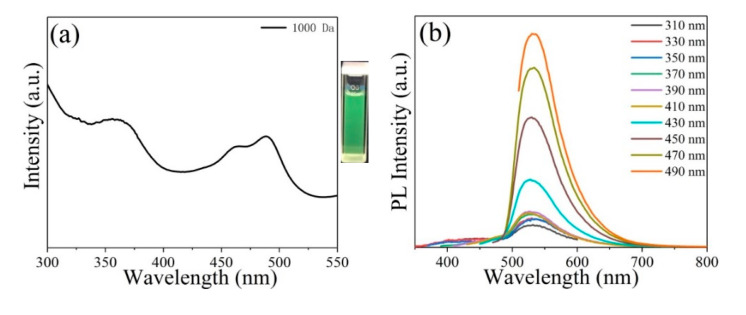
(**a**) UV–VIS absorption spectrum of the GQDs dialyzed with a dialysis bag of 1000 Da molecular weight (inset, optical photographs of the corresponding samples under excitation with a wavelength of 365 nm), (**b**) PL spectra of the GQDs under excitation with a different wavelength, (**c**) PL spectra of the four separated GQDs under excitation at 330 nm, (**d**) PLE spectrum of the GQDs when fixing the emission wavelength at 530 nm.

**Figure 6 molecules-26-03922-f006:**
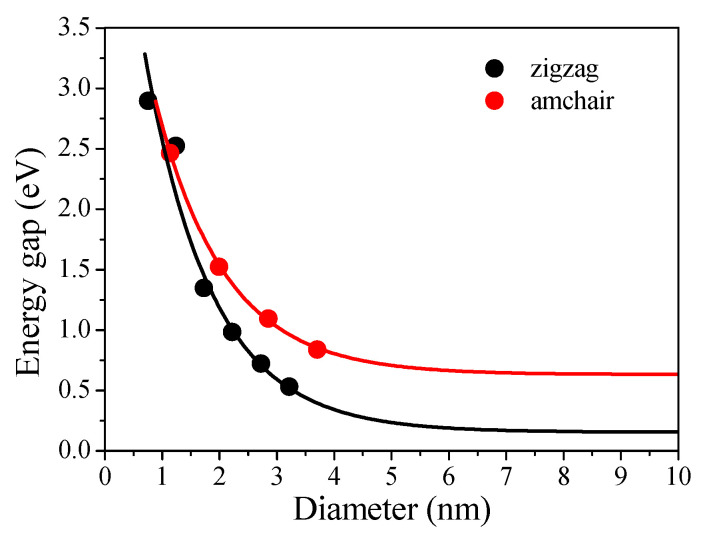
Illustration of the calculated size-dependent electronic gap of the armchair- and zigzag-edged GQDs.

**Table 1 molecules-26-03922-t001:** D and G peak frequencies of the GQD samples obtained from dialysis bags of different molecular weights.

Samples	$D\;\mathrm{Peak}\;\mathrm{Position}\;(cm^{-1})$	$I_{D}$	$G\;\mathrm{Peak}\;\mathrm{Position}\;(cm^{-1})$	$I_{G}$	$I_{D}/I_{G}$
1000 Da	1333.5	89,594.1	1576.61	91,445	0.98
3500 Da	1343.29	37,244.3	1581.99	39,994.2	0.93
7000 Da	1342.23	44,510.4	1575.87	47,180.2	0.94
14,000 Da	1333.78	34,741.8	1572.8	34,708.5	1.00

**Table 2 molecules-26-03922-t002:** QY of the four separated GQDs under excitation at 330 nm using quinine sulfate as a reference.

Samples	Integrated Emission Intensity (I)	Abs. (A)	Refractive Index of Solvent (n)	QY (φ)
Quinine sulfate	15,644	0.109	1.33	0.54 (know)
14,000 Da	397	0.128	1.33	0.02
7000 Da	2321	0.048	1.33	0.18
3500 Da	4651	0.05	1.33	0.35
1000 Da	5047	0.042	1.33	0.45

## Data Availability

The data can be made available upon reasonable request.

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
