# Peer review of "Size Effect of Graphene Quantum Dots on Photoluminescence"

_molecules, 2021, doi:10.3390/molecules26133922_

Round 1
Reviewer 1 Report
This manuscript presents an impressive quantity of spectroscopic results on graphene quantum dots of different sizes. The focus is on photoluminescence intensities, an aspect well worth publishing in Molecules.
The manuscript is well organized and interesting to read.
The authors should better explain the connection between the DFT band gap calculations (Fig. 6) and the observed luminescence spectra(Fig. 5b, 5c). Is the calculated variation of band gap energies (Fig. 6, dots) visible in the spectra in Fig. 5, where the energy of band maxima does not appear to change much? A clarification of this aspect might be considered.
Author Response
Response: Thanks for your recognition of our work. The connection between the DFT band gap calculations and the observed luminescence spectra has been added to the article: “Figure 5c shows that the PL emission peaks of the four separated GQDs are slightly red-shifted. We believe that GQDs of different sizes have different band gap widths, resulting in different PL emission peaks. In order to prove this relationship, we calculated by DFT. The results are illustrated in Fig. 6 With the increasing size of four separated GQDs, the gap of both armchair- and zigzag edged GQDs decreases rapidly at first and will be gentle when the size is greater than 6 nm. When the size is greater than 5 nanometers, the curve starts to become flat. As we all know, the DFT-GGA method often underestimates the band gap, so we assume that whether it is an armchair- or a zigzag edged, when the size is greater than 5nm, the GQD band gap change slightly depends on its size. It can be seen from this that when the size of the GQD is larger, its band gap will become smaller, which is consistent with the experimental results. As shown in Figure 5c, under the excitation at 330nm, the PL emission peaks of the four separate GQDs are slightly red-shifted as the GQD size increases. Because as the size of the GQD increases, its energy gap continues to decrease, and photons are more likely to transition, which results in a red shift of the PL emission peak. This result also confirmed our conjecture.”

Reviewer 2 Report
Liu and his co-workers have synthesized graphene quantum dots (GQDs) using a one-step hydrothermal method, and separated them into four groups with different average particle size using dialysis bags. GQDs were characterized by various techniques, including TEM, XRD, XPS, IR, Raman and UV spectroscopies. The size effect of graphene quantum dots on photoluminescence properties of GQDs was investigated. Experimental results presented in this manuscript are interesting, however, the manuscript is not written well. The English of the text is poor. Authors should consult with a native speaker. Additional comments:
--- 2.1. Synthesis of GQDs:
Line 60: ‘80 mL of nitric acid’, please provide the concentration of the acid, e.g. ‘80 mL of xx% nitric acid’; without concentration the volume does not have any meaning;
Line 61: ‘diluted with distilled water’, provide the volume of the DI;
Line 63: ‘dissolved in a solution of NaOH by ultrasound to adjust the pH value’, provide the concentration and volume of the NaOH solution, as well as the pH value, e.g. ‘dissolved in xx mL xx M aqueous NaOH solution by ultrasound to adjust the pH value to xx’;
Line 70: the sentence does not contain a verb; e.g. ‘was investigated’ is missing; ‘measurements were’ carried out;
--- Chapter ‘2.2. The PL measurement’ should be deleted. The information is provided in chapter ‘2.4. Characterization’;
--- line 91: provide the technique used for recording the FT-IR spectrum (ATR, KBr pellet, etc..??);
--- line 91-92: the following sentence is completely wrong: “Fourier transform infrared spectroscopy (FT-IR) was investigated using a Perkin-Elmer spectrum.” I guess you mean “Fourier transform infrared spectrum (FT-IR) was recorded using a Perkin-Elmer spectrometer.” Please provide the type of this spectrometer and the resolution you used.
--- lines 92-93: provide information about the method/solvent used for recording the UV spectrum (dispersion in water ???);
--- line 94: please replace “carried out” with “recorded”;
--- line 94: please add after the word ‘spectrophotometer’ “at a sample concentration of 0.1 mg·mL-1”;
--- 2.4. Characterization: provide information about the Raman spectrometer and Raman measurement;
--- lines 132-133 and Figure 3c: please elaborate more on the high-resolution O 1s spectrum. Based on C 1s XPS and IR spectra, you expect carboxyl and hydroxyl groups functionalized GQDs. This expectation means three different oxygen environment, and three O 1s XPS bands. Deconvolute the broad and asymmetric O 1s band and analyze accordingly.
Author Response
Thank you very much for your suggestion. We have modified this one by one, see the attachment for details.

Round 2
Reviewer 2 Report
Authors have corrected the manuscript according to reviewers' comments. However, the English of the text is still not good, especially in chapter 2.3. Characterization. Please follow reviewers' comments on the first version of the manuscript.
Author Response
Response: We have done our best to improve the English of the text, especially 2.3 Characterization.
“Transmission Electron Microscope (TEM) was carried out on aused to obtain the mor-phology of the sample by the JEM-2100F (JEOL, Japan) electron microscope. Samples were prepared by placing a drop of dilute aqueous dispersion of GQDs on the surface of a cop-per grid. Powder X-ray diffraction (XRD) spectra were collected on a Rigaku D/MAX 2550 diffractometer within 5°-80° (2 theta). X-ray photoelectron spectroscopy (XPS) measure-ments were made on ESCALAB 250XI with mono Al Kα radiation (hν = 1486.6 eV). Ra-man spectra were recorded on a Laser confocal Raman Spectrometer (Renishaw in-Via) with 514 nm incident radiation. The photoluminescence spectra(PL) and photolumines-cence excitation (PLE) spectra were recorded out using a fluorescence spectrophotometer (Agilent Technologies, Carry Eclipse) at a sample concentration of 0.1 mg·mL-1. UV-vis absorption spectra were recorded on a PerkinElmer Lambda 750 spectrophotometer at a sample concentration of 0.1 mg·mL-1. Fourier transform infrared (FT-IR) spectra were rec-orded using a IRAffinity-1S spectrometer (KBr pellet). Unless otherwise specified, the GQDs (the GQDs dialyzed with dialysis bag of 1000 Da) with the best optical performance were selected for various characterizations.”
